# The Functional Parasitic Worm Secretome: Mapping the Place of *Onchocerca volvulus* Excretory Secretory Products

**DOI:** 10.3390/pathogens9110975

**Published:** 2020-11-23

**Authors:** Luc Vanhamme, Jacob Souopgui, Stephen Ghogomu, Ferdinand Ngale Njume

**Affiliations:** 1Department of Molecular Biology, Institute of Biology and Molecular Medicine, IBMM, Université Libre de Bruxelles, Rue des Professeurs Jeener et Brachet 12, 6041 Gosselies, Belgium; jsouopgu@ulb.ac.be (J.S.); Njume.Ferdinand.Ngale@ulb.be (F.N.N.); 2Molecular and Cell Biology Laboratory, Biotechnology Unit, University of Buea, Buea P.O Box 63, Cameroon; stephen.ghogomu@ubuea.cm

**Keywords:** onchocerciasis, nematodes, excretory-secretory products (ESPs), pathogenicity

## Abstract

Nematodes constitute a very successful phylum, especially in terms of parasitism. Inside their mammalian hosts, parasitic nematodes mainly dwell in the digestive tract (geohelminths) or in the vascular system (filariae). One of their main characteristics is their long sojourn inside the body where they are accessible to the immune system. Several strategies are used by parasites in order to counteract the immune attacks. One of them is the expression of molecules interfering with the function of the immune system. Excretory-secretory products (ESPs) pertain to this category. This is, however, not their only biological function, as they seem also involved in other mechanisms such as pathogenicity or parasitic cycle (molting, for example). We will mainly focus on filariae ESPs with an emphasis on data available regarding *Onchocerca volvulus*, but we will also refer to a few relevant/illustrative examples related to other worm categories when necessary (geohelminth nematodes, trematodes or cestodes). We first present *Onchocerca volvulus*, mainly focusing on the aspects of this organism that seem relevant when it comes to ESPs: life cycle, manifestations of the sickness, immunosuppression, diagnosis and treatment. We then elaborate on the function and use of ESPs in these aspects.

## Highlights

➢As they are parasites, the biology of filariae involves dialogs with their hosts in which ESPs are main actors.➢The immune microenvironment is dynamic across different tissues, and thus, activation of the immune response by nematode ESPs may also be stage- and niche-specific.➢Understanding the evolution of the nematode secretome could improve our understanding of parasitic evolution in nematodes.➢Addressing the function of nematode ESPs remains crucial for a proper understanding of host–parasite interactions.

## 1. Introduction

### 1.1. The Parasitic Way of Life

According to the encyclopedia Britannica (Britannica.com), parasitism is a “relationship between two species of plants or animals in which one benefits at the expense of the other, sometimes without killing the host organism”. According to the Cambridge dictionary, it is the phenomenon whereby one animal or plant living on or in another animal or plant of a different type feeds on it; it therefore involves at least two organisms, a parasite and a host. As life has been invading most known biotopes and environments available on Earth (from the ice of the poles to near-boiling water in the geysers or at higher temperatures in the seafloor hydrothermal vents), it would have been a missed opportunity not to take living organisms themselves as life support. Parasitism is indeed a very common way of life, as it has been estimated that most organisms on Earth are parasites at least at one stage of their life cycle [1]. Why? An evident answer to this question is that a parasite lives on or inside a food source and does not therefore have to consume energy to look for its food. This is perhaps best-illustrated by blood stream parasites—for example African trypanosomes—living in an ocean of nutrients, as they are inside the tissue in charge of bringing nutrients to other tissues of the organism.

Nevertheless, the parasitic way of life also has many drawbacks or requirements: The parasite is dependent on its host for a living and has therefore to be transferred to another host before the previous one dies. It has to develop ways to communicate with its host, as it is its sole source of nourishment. As most living organisms have evolved, though to different extents, defense mechanisms towards intruders—for example, the immune system—parasites have to develop means to counteract or go around these defenses in order to have a successful stay in their host. As a consequence of these properties, many parasites are detrimental to their hosts. Amongst other things, they consume the energy of the host and interfere with their integrity, thereby becoming pathologic agents responsible for many diseases.

As already said, parasites are very common. It seems, however, that some phyla or categories are particularly prone to the emergence of parasitism. Among them are viruses (that are parasites by nature), prokaryotes, protozoan and worms. Several phyla or classes of the latter, the nematodes (N throughout the text), Platyhelminthes cestodes (C throughout the text) and trematodes (T throughout the text) are responsible for some of the most widespread human parasitic pathologies on Earth. They are collectively responsible for more than 14 million disability-adjusted life years (DALYs) [2]. However, when it comes to the number of infection cases, the champions among worms are nematode geohelminths (*Strongiloides*, *Ancylostoma*, *Necator*, *Ascaris*, *Trichuris* and *Enterobius* spp.), contributing to the global burden of 1.5 billion people infected with geohelminths, according to the WHO (https://www.who.int/news-room/fact-sheets/detail/soil-transmitted-helminth-infections), and filariae (*Wuchereria*, *Brugia*, *Onchocerca* or *Mansonella* spp.), contributing to a total of 120 million worldwide infections for filariasis, according to the WHO [3].

In their struggle for survival, parasites employ a panoply of strategies. Intracellular parasites hide from the immune system of their host. Other parasites remain exposed but rely on antigenic variation to evade host immune responses. Yet other strategies may involve molecular mimicry, a molecular disguise to look like the host and mislead the immune system. As opposed to protozoans and prokaryotes, nematodes and other helminth parasites do not usually replicate in their host, and adult individuals have to stay exposed to a functional immune system for years. Thus, these categories of parasites function on a different strategic platform to survive in their host, often requiring the release of molecules to modulate, suppress or mimic host immune effector mechanisms. The amount released could be so important that it could act as a smoke screen, hijacking the vigilance of the immune reaction. The toolset involved in these tasks is mainly comprised of the excretory-secretory products (ESPs) of the worm. In this review, we delve into analyzing the molecular actions and functions of nematode ESPs and their potential applications in disease management of, mainly, filarial but, also, when relevant, nonfilarial diseases.

### 1.2. Onchocerciasis

*Onchocerca volvulus* is a nematode worm. Nematodes constitute an important (in terms of species number) phylum [4] where recurrent independent events of the evolution to parasitism occurred [5], a propensity still not explained [6,7]. Several members of nematode clade III [8], including *Onchocerca volvulus*, are responsible for lymphatic filariasis in humans with tremendous health and socioeconomic consequences [9]. According to regular nematode features, the life cycle of *Onchocerca volvulus* goes through four larval and an adult stage. As seen from the patient’s point of view, the life cycle of *Onchocerca volvulus* begins with the transmission of the infective L3 stage into the human blood during an infected blackfly’s blood meal. The L3 penetrates the bite wound, molts to the L4 stage in the subcutaneous tissues and further develops into either an adult male or adult female (Figure 1). The adult female worms eventually become encapsulated in nodules, where they remain for the rest of their up to 15-years’ lifespan, whereas the males move in subcutaneous spaces in and out of nodules to fertilize the stationary females. *Onchocerca volvulus* worms are ovoviviparous; fertilized females release an average of about 1500 unsheathed microfilariae (mf) per day [10] that migrate through the skin where they can be picked up again by a blackfly upon another blood meal. The mf migrates to the thoracic musculature of the fly, where it develops through the L1 stage to the L2 and L3 stages. The L3 stage then migrates to the proboscis of the fly, ready for transmission to the human host upon a blood meal.

The global disease burden of human onchocerciasis was recently estimated at 20.9 million infected people (99% of them living in sub-Saharan Africa). The sickness is associated to symptoms including mostly skin diseases and eye lesions culminating in blindness but, also, lymphadenitis and epilepsy. Clinical lesions are believed to be due to inflammatory cascades triggered by dying mf releasing their *Wolbachia* endosymbiotic bacteria [11].

The diagnosis of the disease still poses a major hindrance to its management. The skin snip test, of simple use, remains the gold standard for diagnosis. It is specific but poorly sensitive. This is probably one of the reasons why it still falls short of assessing control programs. It is also invasive and, therefore, associated with potential infection problems. Therefore, various other types of assays have been developed for diagnosis of the disease, including antibody capture assays using recombinant antigens such as Ov16 [12], Ov20 and Ov33 [13]; a cocktail of four recombinant antigens Ov-FAR-1, Ov-API-1, Ov-MSA-1 and Ov-CPI-1 [14] and many others, as reviewed in Table 1. Antigen capture assays have also been attempted [15,16,17,18]. Other approaches, including a PCR-based diagnosis [19], a metabolomics-based approach [20] and liquid chromatography-tandem mass spectrometry [21], have been reported as well. Nowadays, only an Ov16 antibody capture test is used for sero-surveillance of the disease [13,17,18,22,23]. The latter, as all other tests, has various drawbacks, including its inability to distinguish between past and active infections and lack of balance between sensitivity and specificity, as well as inappropriate field applicability and cost.

The treatment of the disease is based on the administration of ivermectin [24], which is insufficient, even more so as there is increasing evidence suggesting the emergence of resistance to this elimination tool [25,26,27].

As discussed above, *Onchocerca* antigens have already been characterized for vaccines and diagnostic and therapeutic purposes (Table 1). As an example, one of the best-characterized *Onchocerca volvulus* diagnostic antigen so far is Ov16. The use of Ov16 as a possible marker for infection began with Lobos et al. in 1990, when they screened λgt11 complementary DNA (cDNA) expression libraries from mf-producing female *Onchocerca volvulus* worms using an affinity-purified antibody from West African onchocerciasis patients. In this study, with the aid of immunoelectron microscopy, the protein was localized in the hypodermis, cuticle and uterus. In 1991, Lobos et al. proposed the antigen to be an early marker of infection [23]. So far, no functional studies have been carried out on the Ov16 antigen, but the INTERPRO analysis indicates it has a phosphatidylethanolamine-binding domain.
pathogens-09-00975-t001_Table 1Table 1A list of published antigens of *Onchocerca volvulus* and their potential applications and functions. ESP: excretory-secretory product and MIF: migration inhibitory factor.Published AntigensProposed ApplicationFunction DomainProtein CategoryPublicationOv-RAL-2/Ov-17ProtectiveDomain of Unknown FunctionPredicted ESP[28]Ov-7, Ov-CPI-1, Ov-CPI-2, OC9.3, OnchocystatinProtectiveCystatinESP[29,30]Ov-1CFPotentially DiagnosticIntermediate filamentESP[31,32]OvSOD1Potentially ProtectiveSuperoxide dismutase 1Cytosolic[33,34]Ov-20, Ov-FAR-1, OvMBP/11ChemotherapeuticFatty acid and retinol bindingESP[35]Ov103, OvMSA-1ProtectiveUncharacterized conserved proteinESP[36,37]Ov-9M/Ov-CAL-1ProtectiveCalponin-like proteinCytosolic[38,39]Ov-FBA-1ProtectiveAldolaseIntracellular[40]Ov-ENOPotentially protectiveEnolaseIntracellular[41]Ov-16DiagnosticPhosphatidylethanolamine binding proteinESP[23]Ov-33, Ov-API-1, OC3.6DiagnosticAspartic Protease InhibitorESP[42]OvB20Potentially protectiveNo functional annotationIntracellular[43]MOv-14, OvTrop, Ov-TMY-1ProtectiveTropomyosinIntracellular[44]Ov-GST-1Potentially chemotherapeutic; Potentially protectiveGlutathione S-transferaseESP[45]M3, M4Potentially Chemotherapeuticn-acetylcholine receptor subunitTransmembrane[46]OvRAL-1Potentially protectiveCalreticulinESP[47]Ov-ALT-1Potentially protectiveSecreted Larval acid proteinsESP[48]Ov-ASP-1ProtectiveActivation-associated secreted protein 1ESP[49,50]Ov-CHI-1, Ov-CHI-2ProtectiveChitinaseESP[51,52]Ov-B8ProtectiveRemodeling and spacing factor 1Intracellular[39]OvMSP-2DiagnosticMajor Sperm Protein 2Intracellular[53]Ov58GPCRPotentially DiagnosticGPCR * (Intimal thickness-related receptor)Transmembrane[54]Ov-MIF-1, Ov-MIF-2ImmunomodulatoryMacrophage Migration Inhibitory FactorESP[55]OvGM2APDiagnostic/Potentially ImmunomodulatoryGM2 activator proteinESP[56]OvMANE-1DiagnosticChimeric antigenNot defined[57]CyclophilinTherapeuticPeptidyl-prolyl *cis*-*trans* isomerasesIntracellular[58]* G protein coupled receptor.


## 2. Excretory Secretory Products (ESPs)

As already said, one of the major requirements of the parasitic way of life is the capability to communicate with the host. This involves the ability to acquire growth factors or nutrients through receptors, channels or engulfment mechanisms—like phagocytosis, for example. It also involves signaling in the other direction, from the parasite to the host, which can be done by any surface molecule but is easier to conceive as performed through mobile secreted molecules. Mediators of these contacts include the ESPs. The term ESP is a generic word for products released to the external environment of the parasite. It encompasses both active secretions via the classical secretory pathway (involving a signal peptide) or alternative pathways (exosomes, lysosomes and microvesicles) from secretory tissues, as well as releases of digestive enzymes from the intestine and secretions from the uterus during egg laying (which might play extra roles in the external environment additional to their canonical functions within the worm). These ESPs include all kinds of biomolecules: proteins, glycoproteins, glycolipids and polysaccharides. A rational approach of the interspecific communication thus involves an analysis of all ESPs, irrespective of their physiological origin, and to subject them to a full range of biochemical, immunological and proteomic analyses. We will review the possible roles and functions of ESPs, focusing on filariae and, more precisely, *Onchocerca* but, also, referring to other worms when relevant to *Onchocerca volvulus.* We will also review their potential for diagnostic, vaccine or therapeutic relevance.

### 2.1. Immunomodulation: Helminths Excretory Secretory Products and Interaction with the Immune System

As stated above, one of the major requirements of the parasitic way of life, essential for parasite survival, is the capability to counteract the immune defenses. This ability has probably been built during the very long coevolution between host and parasite, often compared to an arms race. One of the host weapons is the immune system, while several types of countermeasures have been acquired by different parasites, including burrowing in the host’s cells, antigenic variation, molecular mimicry, smoke screen or immunosuppression. The ability to escape the immune response is particularly critical for helminths, which survive for years (sometimes more than a dozen years) in their host under constant surveillance of the immune system. ESPs (and surface antigens) probably play a central role in signaling to the immune system, since, as multicellular organisms, they cannot be phagocytosed but, yet, are capable of eliciting an adaptive immune response.

Extracellular membrane vesicles (EVs) have only recently been characterized. As evident from their name, they are acellular structures surrounded by a phospholipid membrane. They fall into several categories generated by different mechanisms: apoptotic bodies (produced during apoptosis), microvesicles (budding of the plasma membrane) and exosomes (exocytosis of multivesicular bodies). They are produced by all kinds of organisms, including worms and, by definition, are part of the ESP realm. They are very complex, as they potentially contain all classes of biomolecules, including nucleic acids, and are believed to be important actors in intercellular communication, as they can transfer signaling or controlling molecules—for example, miRNA. There is evidence for the uptake of extracellular vesicles by host cells from various protozoans and, also, worms, including human parasites *Fasciola hepatica* (*T*) and *Schistosoma mansoni* (*T*) or animal parasites *Echinostoma caproni* (*T*), *Ascaris suum* (*N*), *Heligmosomoides polygyrus* (*N*), *Trichuris suis* (*N*) and *Trichuris muris* (*N*) [59,60,61,62,63,64]. Nevertheless, the demonstration of microvesicles from human parasitic nematodes (and, therefore, filariae) active in immunosuppression still awaits evidence, although there are some promising data in animal models [65].

One of the observed recurrent features of infections by helminths is a Th2 skewed immune response, as testified by the production of Th2 diagnostic cytokines (typically interleukin (IL)-4, IL-5 and IL-13) and immunoglobulin G4 (IgG4) antibodies. Although the canonical immune response to helminth infections is biased to Th2 in nature [66,67], some exceptions have been noted where an initial Th1 response is mounted that later changes to a Th2 response as the infection progresses [68,69]. Helminths can also dampen the immune response by expanding regulatory T cells indirectly through dendritic cells (DCs) [70,71], as well as acting directly on naïve T cells using a transforming growth factor-beta (TGF-β) mimic [65]. Thus, the Th1–Th2 switch could involve both dampening an original Th1 response, reducing tumor necrosis factor-alpha (TNF-α) and interferon gamma (INFγ) expression or triggering a Th2 phenotype. All actions have indeed been experimentally observed and some of the actors identified.

Some parasites have organelles dedicated to the production of ESP excretion and/or secretion. This is the case of the protozoan *Toxoplasma gondii*. The apical complex gathers several vesicular organelles—namely, micronemes, rhoptries, dense granules and exonemes—all dedicated to the secretion of dozens of molecules implicated in the intracellular lifestyle of this protozoan parasite [72,73]. A typical illustration is given by molecules such as ROP16, a kinase able to interfere with the phosphorylation of STAT transcription factors involved in the Th1/Th2 balance [74], or the triad ROP5, 17 and 18 kinases and pseudo-kinases able to interfere with the phosphorylation of immunity related GTPases (IRGs), essential actors of the anti-*Toxoplasma gondii* innate immune response [75,76]. However, as *Toxoplasma gondii* is not a nematode worm, we will not develop those examples further.

Helminth (ESPs) can have a direct effect on both the innate and adaptive immunity. An intruder indeed first elicits a quick but nonspecific so-called innate immune response. This response is mainly triggered by the stimulation of pattern recognition receptors by PAMPs (pathogen-associated molecular patterns) widely expressed by pathogens. The initiating step in the adaptive immune response involves antigen presentation by an antigen-presenting cell (APC)—usually, a DC. DCs upregulate the expression of surface ligands and soluble mediators, which activate antigen-specific T cells through their costimulatory and cytokine receptors. This interaction can direct the qualitative nature of the response towards a dominant Th1 or Th2 mode [77] into the regulatory pathway of FoxP3+ T-regulatory cells (Tregs) [78] or to alternative modes that continue to be uncovered. In the context of helminth infections, DCs exhibit a very specific phenotype. As a general rule, they fail to classically mature in that they do not upregulate coreceptors, nor do they release proinflammatory cytokines [79]. This phenomenon seems to be common to most helminth infections, irrespective of their taxonomic classification (cestode, trematode or nematode). The simplest explanation for this observation is that worms send signals that allow APCs to drive the Th2 phenotype and/or inhibit the Th1/Th17 phenotype [80].

Some of the best-characterized actions of ESPs are from nonfilarial nematodes. *Schistosoma mansoni* and *Fasciola hepatica*, both trematodes, peroxiredoxin orthologs alternatively activate macrophages and are instrumental in promoting parasite-induced Th2-type immunity [81]. DCs exposed to ESPs derived from *Taenia crassiceps*, a cestode tapeworm, fail to upregulate CD83, HLA-DR, CD80 or CD86 [82]. This immature phenotype persists even when these DCs are subsequently stimulated with lipopolysaccharides (LPS). *Echinococcus granulosus* (also a cestode) also bias the immune reaction to a regulatory state. This action is mediated through sheep hydatid fluid (SHF) and antigen B (AgB), which suppress monocyte differentiation and DC maturation, as assessed by the dysregulation of costimulatory molecules and inflammatory cytokines [83]. The secreted thioredoxin peroxidase of *Echinococcus granulosus* (*C*) was recently shown to alternatively activate macrophages through the PI3K/AKT/mTOR pathway [84]. An experimental administration of ESPs from the nematode *Nippostrongylus brasiliensis* induced a Th2 response strong enough to counteract the Th1-polarizing Freund’s adjuvant [85]. These same observations have been reported with other helminth ESPs, including the *Schistosoma mansoni* soluble egg antigen (SEA) [86] that skews the response to Th2, polarizes DC accordingly and inhibits Toll-like receptor (TLR) stimulation [87]. In the nematode *Trichinella spiralis*, ESPs were found to suppress in-vitro DC maturation in a TLR4-dependent manner and, also, induce the expansion of Tregs [88].

DCs exposed to a wide array of helminth ESPs have been shown to promote the expansion of CD4+CD25+Foxp3+ regulatory T cells [88,89,90,91]. Additionally, DCs matured in the presence of *S. mansoni* phosphatidylserine seem to acquire the capacity to potently drive naïve T cells to become regulatory in nature [92], and the nematodes *Trichinella spiralis* and *Heligmosomoides polygyrus* stimulate the expansion of existing Treg cell populations in vitro and ex vivo, respectively [88,93]. DC conditioning by ESP also results in the release of regulatory cytokines such as IL-4 and IL-10. These cytokines are most commonly produced either by DC-stimulated T cells or by DCs themselves [94]. Omega-1 (an RNase T2 binding to mannose receptors through its glycan moieties), an abundant ESP of *Schistosoma mansoni* (*T*) eggs, is able to induce the differentiation of naïve T cells into FoxP3+ T cells in a non obese diabetic(NOD) mice model [95].

Some molecular players of immunomodulation have also been described for some few filariae. An ESP from *Acanthocheilonema vitae* (*N*), ES62, displays several abilities also shared by other ESPs from other helminth species. ES62 can prime DCs towards a Th2 phenotype and inhibit Th1 and Th17 responses [96]. Furthermore, ES62, as well as Lactose-N-fucopentaose III (LNFPIII) and the *Dirofilaria immitis* (*N*)-derived antigen (DiAg), can induce secretion of the anti-inflammatory cytokine IL-10 [97,98,99]. The macrophage migration inhibitory factor (MIF) of *Brugia malayi* (*N*) can trigger the differentiation of macrophages to an alternative M2 phenotype [100]. Additionally, in *Brugia malayi* (*N*), ESPs from live mf have been shown to induce DC death and inhibit the production of IL-12 and IL-10 [90]. MIF from other nematodes, including *Ancylostoma ceylanicum* [101], *Onchocerca volvulus* [55], *Wuchereria bancrofti* [102], *Strongyloides ratti* [103], *Haemonchus contortus* [104], *Ostertagia ostertagi* [105] and *Meloidogyne incognita* [106], have also been reported to play various roles in the modification of host immune responses. Although peroxiredoxins from other helminths such as *Shistosoma mansoni* (*T*) and *Fasciola hepatica* (*T*) have been reported to alternatively activate macrophages [81], peroxiredoxin from the parasitic nematode *Teladorsagia circumcincta* could not alternatively activate murine macrophages in vitro [107]. The asparagyl transfer RNA synthetase of *Brugia malayi* (*N*) has been reported to have potent anti-inflammatory properties in vivo, strong enough to resolve the gut pathology in T-cell transfer mouse models of colitis [108].

Extracellular vesicles (EV) could, of course, exert various biological activities in the context of worm infections in terms of immunosuppression. This is still poorly characterized. It is, however, illustrated by their ability to suppress the expression of proinflammatory cytokines such as IL6, TNF-α and IL17a [109,110].

### 2.2. Molecular Pathways Involved in ESP Actions

The signal transduction pathways involved in ESP actions are poorly known. However, the data illustrate several ways to drive host immunity toward hypo-responsiveness and/or immunological tolerance. In some cases, the molecular basis has been uncovered, which often involves pattern recognition receptors (PRR). These are sensors, such as TLRs (Toll-like receptors), NLRs (NOD-like receptors—for example, the inflammasome) or CLRs (C-type lectin receptors), widely expressed by APCs such as DCs and macrophages, triggering the innate immunity response. These receptors recognize molecular patterns that are conserved and widespread amongst pathogens. Some ESPs are indeed post-translationally modified by moieties that are ideal candidates as ligand for some PRRs or are themselves biomolecules bound by PRRs, such as lysophosphatidylserine, sugars or RNA. For example, DC-SIGN (a CLR) plays a critical role in the recognition of *Brugia malayi* (*N*) but, also, *Toxocara canis* (*N*) and *Fasciola hepatica* (*T*) [111,112,113].

*Trichuris suis* (*N*) is a rich source of prostaglandin E2 (PGE2), a lipid-based molecule presumably binding TLR4, a possible mechanism for the observed reduction of MYD88 and TRIF signaling [114,115]. *Acanthocheilonema viteae* (*N*) ES62’s action depends on a phosphorylcholine post-translational modification [116,117] that could mediate interactions with TLR4, as TLR4 or MYD88-deficient DC do not respond to ES62 exposure [118]. *Brugia malayi* (*N*) microfilariae cocultured with DC interfere with mTOR phosphorylation and induced autophagy, as testified by beclin1 phosphorylation, LC3II induction and p62 degradation [119]. LNFPIII is post-translationally modified by fucose addition that probably mediates its action through TLR4 [120] and could also act through the type C lectin receptor [98].

Together with the activation of the NF-κB transcription factor pathway, the MAPK pathway is one of the main signal transduction pathways elicited by TLRs leading to transcription factor activation, the expression of molecules involved in APC maturation, such as costimulatory molecules, major histocompatibility complex(MHC)and cytokines. From the data available, it seems that, of the three MAPKs, ESPs of nematodes preferentially activate ERK rather than p38 and JNK. As examples, ES62 activates the ERK pathway in macrophages [121]. LNFPIII was reported to induce the sustained activation of ERK but not p38 and JNK activation in DCs [120]. *Acanthocheilonema viteae* (N) cystatin also activates the ERK pathway but not the JNK pathway [122]. It is also known to activate the p38 pathway.

Some ESP actions have also been documented that are not TLR-mediated. The first mechanism interferes with antigen presentation and, thus, T lymphocyte (TL) priming. Antigen presentation indeed first involves the chopping of the antigen into peptides, an action ensured by proteases. Interfering with their actions (through protease inhibitors) is a way to downregulate antigen presentation, and this way has been taken by several nematodes. The *Brugia malayi* (*N*) BmCPI-1 (cysteine proteinase inhibitor-1) and BmCPI-2 interfere with cysteine protease function in APC and, thereby, with antigen presentation and, thus, TL priming [123]. A novel cystatin of *Trichinella spiralis* (*N*) has also been shown to interfere with antigen presentation by the depletion of MHC class II expression [124]. The *Onchocerca volvulus* cystatin onchocystatin reduces HLA-DR expression [125]. Indirect evidence suggests that the *Acanthocheilonema viteae* (*N*) cystatin [126] can further act through scavenger receptors and TGF-β receptors, thereby polarizing the response to Th2.

Further suggestions involve the promotion of worm longevity, as observed in *Nippostrongylus brasiliensis* (*N*), where the infection can be confined by an anti-cystatin antibody, indicating a possible role of cystatins in restricting antigen-loading and presentation by MHCII [127]. Omega-1 colocalizes with the cytoskeleton, interfering with DC-TL intercellular contact and TCR signaling [128]. ESPs also interfere with protein kinase C (PKC) signal transduction. ES-62 promotes PKC degradation [96,129,130], an essential component in the downstream signaling by T-cell receptors and B-cell receptors, thereby restricting cell proliferation and antibody production. In the context of allergic reactions, ES-62 promotes PKCα or PKCδ sequestration and degradation in mast cells, thereby inhibiting FcεI-mediated activation by IgE [131]. DiAg also impairs IgE-mediated degranulation [132].

ESPs can also interfere indirectly with signal transduction through molecules that are regulators of signal transduction pathways, such as SOCS3, which controls TLR signaling [133,134]. *Brugia malayi* (*N*) ALT2 (abundant larval transcript 2) triggers GATA-3 and SOCS-1 expression, thereby promoting Th2-type and attenuating IFNγ-dependent signaling [135]. Cystatins are protease inhibitors that have been documented as ESPs of most analyzed filarial nematodes [136]. *Acanthocheilonema viteae* (*N*) cystatin is internalized by macrophages and stimulates DUSPS (dual-specific phosphatases), which, in turn, interfere with MAPK signaling pathways and the phosphorylation of CREB and STAT3 transcription factors. Thus, they induce an Mreg phenotype, which, in turn, suppresses DC activation through interleukin secretion and cell–cell contact [122,137]. *Acanthocheilonema viteae* (*N*) is able to interfere with another TLR-elicited pathway—namely, the PI3K—also inducing DC maturation [122,138].

*Brugia malayi* (*N*), but, also, *Heligmosomoides polygyrus* (*N*), *Fasciola hepatica* (*T*), *Shistosoma mansoni* (*T*) and *Ancylostoma caninum* (*N*) [139], secrete molecules mimicking TGF-β homologs (BmTGH-1 and 2), themselves able to mimic TGF-β-elicited effects. In the case of *Heligmosomoides polygyrus* (*N*) HpTGM, it has been shown to bind the TGF-β receptor and induce Foxp3+ Treg cells in mice and humans [65].

*Brugia malayi* (*N*) [140], but, also, *Trichuris suis* (*N*), *Shistosoma mansoni* (*T*) and *Taenia taeniaeformis* (*C*) [141,142], secrete molecules mimicking PGE2, known for its capacity to shift immune responses towards the Th2 phenotype. *Brugia malayi* (*N*) BmK1 and *Ancylostoma Ceylanicum* (*N*) AcK1 are able to block T-cell Kv1.3 channels essential for T-cell activation [143].

In summary, there are only very few examples of filariae ESP with characterized functional and/or molecular mechanisms interfering with the immune system of their host. These examples clearly illustrate the fragmented or piecemeal state of our knowledge. Nevertheless, they clearly explain the observed capability of the worms to interfere with the immune response of their hosts.

### 2.3. ESPs and Parasitic Evolution

So far, the attempts to understand the evolution of parasitism in nematodes have largely relied on a comparative genomics/proteomics/transcriptomics approach between free-living nematodes or nematode stages vs. parasitic nematodes (or nematode stages) attempting at identifying a potential expansion or compression of gene sets that could be related to parasitism. Genes involved in cuticle biogenesis/formation have been targeted, but the expansion of the parasitic collagen family, for example, does not necessarily correlate with parasitism, though suggestive of an adaptation of the cuticle to a parasitic lifestyle. Evidence supporting the Dauer stage (a developmentally arrested stage) of free-living nematodes as the evolutionary precursor of the infective larvae initiating parasitic life in their mammalian host has been reviewed [144]. However, increased molecular characterization comparing gene products involved in Dauer formation and infective larvae formation, such as the daf-12 nuclear receptor, as well as the daf-7-TGF-β ortholog, would be necessary to throw more light onto the evolution of nematode parasitism. Parasitic orthologs of TGF-β have been reported in ESPs, and a secreted TGF-β mimic of *Heligmosomoides polygyrus* (*N*) was reported to have a direct effect on the TGF-β receptor, inducing the expansion of Foxp3+ Treg cells [65]. This parasite TGF-β mimic showed no sequence similarity to the mammalian TGF-β but could induce the expansion of Foxp3+ Treg cells more potently than the human TGF-β, clearly suggesting a convergence of function.

A good nematode model to address the question of parasitic evolution is the *Strongyloides* genus, where species have both free-living and parasitic stages. In *Strongyloides* spp. (*N*), two key ESP gene families (SCP/TAPs) and astacins were found to be predominantly expanded in the parasitic adult females compared to free-living females [145]. Additionally, about 43% of the genes upregulated in the transcriptome of parasitic females were predicted to encode a signal peptide compared to 26% for the free-living females, suggesting an expansion of the pool of secreted proteins. In *Angiostrongylus cantonensis* (*N*), an expansion of the astacin family was observed compared to *Caenorhabditis elegans* (*N*) [146]. This, therefore, supports the assertion that ESPs are an important arm in the selection drive towards parasitism. The argument that gene expansion or compression could be indicative of parasitic evolution is still debatable, as, while some gene families are expanded, some parasitic species are known to have a smaller genome size compared to their free-living counterparts. An example is the case of the nematode hookworm *Necator americanus*, which is known to have about 5000 less genes compared to the free-living *Caenorhabditis elegans* (*N*). Nevertheless, a decrease in genome size (perhaps related to the disappearance of some metabolism branches whose products can be supplied by the host) does not preclude the specific expansion of some other gene families.

Surface proteins lining the cuticle of nematodes have been reported to be species-specific, as well as stage-specific [147,148,149,150,151]. Surface proteins of filarial nematodes do not, however, appear to be as specific as in the other classes of nematodes. Although these surface proteins are glycoproteins, a leitmotif in surface proteins, they differ from other glycoproteins in that their carbohydrate moieties are buried within the cuticle rather than being exposed, as is the case with other membrane glycoproteins that form the glycocalyx [152]. An alternative way to access the evolution of nematode parasitism will be to analyze the evolution of the secretome between free-living and parasitic species/stages. The evolution of the nematode secretome has been seldom addressed, and this remains crucial, as it should contain the toolbox required for the survival of nematodes within their diverse niches. Accordingly, an increasing number of studies are being directed at the secretome of nematodes, as some have been described for *Nippostrongylus brasiliensis* [153], *Necator americanus* [154] and *Ancylostoma caninum* [155]. A description of the ES proteomes across the various nematode clades will throw more light onto the evolution of the nematode secretome.

Another study involving entomopathogenic nematodes of the genus *Steinernema* identified a set of ESP genes involved in host-seeking behaviors but not active infection [156]. This study focused on a set of 52 conserved ESP genes between *Steinernema feltiae* and *Steinernema carpocapsae*, two species with varying host ranges and specificity. *Steinernema* infective juveniles (IJ) are considered to be the equivalent of the *Caenorhabditis elegans* (*N*) Dauer and are the only free-living stage in the *Steinernema* life cycle. The study suggests that the gene set activated before IJ becomes parasitic and, during active infection, constitutes the gene set that promotes its parasitic way of life. In the aforementioned study, the in-vitro and in-vivo ESP profile of *Steinernema* were found to be asymmetrical, indicating that the true tool set involved in immunosuppression/immunomodulation might not be conclusive if information on the ESP profile is gathered exclusively from the in-vitro ESP profile.

Although there is a general assumption that parasitism evolved from free-living nematodes, it remains debatable whether some free-living forms observed in our time could have evolved from parasitic species. An answer to this question could again be provided by a genomic analysis. As can be guessed from the preceding examples, most of the data gathered on nematode evolution towards parasitism have been harvested on geohelminths, and a lot remains to be done to address the evolution of filariae.

### 2.4. ESPs in Neurologic Diseases

A considerable number of nematodes possess visas that permit them to penetrate the central nervous system (CNS) of their hosts. Of all these nematodes, true residency in the CNS is a peculiarity of *Angiostrongylus cantonensis*, a nematode affecting rats and humans in the tropics. For *Angiostrongylus cantonensis* (*N*), the larval stages are known to migrate through the blood–brain barrier (BBB) and enter the CNS, where they cause pathology [157,158]. Mechanistically, plasminogen activators and matrix metalloproteinases have been implicated in the disruption of the BBB in mice infected with *Angiostrongylus cantonensis* (*N*) [159]. More generally, metalloproteinases have been reported to be constituents of nematode ESPs and associated with reduced infectivity [160,161].

For those species who are believed to be occasional visitors to the CNS, such as *Onchocerca volvulus*, the mechanisms by which their presence associates with neurologic diseases remain unclear. In fact, the *Onchocerca volvulus* association with epilepsy is still debatable, as not all studies have found a correlation [162,163]. Many different mechanisms have been proposed for the pathogenesis of onchocerciasis-related epilepsy: mfs invading the brain, the humoral immune response, sleep deprivation due to intense itching [164], *Onchocerca volvulus* strain specificity, concomitant infections such as measles, genetic predisposition or malnutrition are all considered possible causes of onchocerciasis-related epilepsy [164,165,166]. The *Wolbachia* endosymbiont of *Onchocerca volvulus* has also been proposed as a possible cause of nodding syndrome and onchocerciasis-related epilepsy [167]. One of the rare molecular tracks regarding the mechanism is provided by the observation that affinity-purified leiomodin-1 antibodies cross-react with tropomyosin, as well as several other proteins in the *Onchocerca volvulus* lysate, implicating autoimmunity as a possible disease-causing mechanism [168].

As ESPs could cross the BBB more easily than a whole worm, the role of ESPs in the pathogenesis of neurologic diseases is a good bet, yet remains to be investigated. ESPs indeed contain proteases and other enzymes that aid in host tissue penetration and could possibly themselves penetrate into the CNS and initiate pathology. Other ESPs unrelated to proteases could be neurotoxic. Alternatively, the immune response against ESP proteins may cause damage to the BBB and provide parasites with access to the CNS. Astacins, a class of ESP proteases known to play roles in molting and tissue penetration, was found to be expanded in *Angiostrongylus cantonensis* (*N*) compared to the free-living *C. elegans* and could confer in *Angiostrongylus cantonensis* (*N*) the ability to penetrate the CNS. This possibility of ESPs aiding parasites to migrate to the CNS is increasingly likely based on observations from the *Angiostrongylus cantonensis* (*N*) model where cytokine secretion was induced in astrocytes following the treatment with ESPs of *Angiostrongylus cantonensis* (*N*) young adult worms [169]. Additionally, ESPs of *Angiostrongylus cantonensis* (*N*) larvae have been shown to induce autophagy in astrocytes [170].

### 2.5. ESPs and Wound Healing

Nematodes can infect their host using different portals, such as the skin, the intestines, the lungs and the brain. In order to get to their final destination, they also have to break through epithelial barriers. These different crossings probably involve proteases in the ESPs. However, it is not usually in their interest to leave these barriers open, as bacteria will follow through and evoke an even stronger Th1 response that could be detrimental to the worm. Indeed, according to the best interests of the worms, their secretory arsenal contains molecules that could aid the rapid healing of the wounds they create as they burrow through epithelial barriers. The involvement of nematodes in wound healing is two-pronged; firstly, the immune response generated in response to nematode infections tends to be Th2-biased, which is also the arm of the immune response responsible for wound healing. Although the activation mechanisms of wound healing are largely based on the damage as a result of the size of the intruding nematodes, antigens in the ESPs are also considered as additional initiators of the Th2 response [95,171]. Molecular initiators of the process will therefore be suitable wound-healing candidates, and ESP molecules are candidates to the initiation process. Secondly, ESPs could actively be involved in wound healing. Ov-GRN-1, a secreted protein of the carcinogenic human liver fluke *Opisthorchis viverrine* (*T*), has been reported to play a role in angiogenesis and wound healing [172]. Ov-GRN-1 has orthologs in nematodes, as well, although the function in the nematodes is yet to be ascertained. It is entirely possible that nematode ESPs do contain other mediators that have a direct impact on the epithelial sites they breach during infection.

### 2.6. ESPs and Worm Cuticle

Although the nematode cuticle is largely composed of proteins such as collagens and cuticlins, the surface coat also contains glycoproteins. The description of a functional cuticle has been largely based on the former proteins, as they are considered essential for cuticular integrity. However, ESPs could also play essential roles in ensuring cuticle integrity. As an example, GM2AP is a nematode ESP that was recently characterized in *Onchocerca volvulus* [56]. The GM2-activator protein was recently shown, for the first time in a nematode species, to be essential for cuticular integrity in *Caenorhabditis elegans* (*N*) (Njume et al., in preparation). The *Onchocerca volvulus* orthologous protein, sharing 65% amino acid identity with the *C. elegans* protein, failed to rescue the phenotype in *C. elegans* GM2AP knockout strains. The lack of rescue with the *Onchocerca volvulus* protein despite the high amino acid identity with the *Caenorhabditis elegans* (*N*) protein suggests that subtle functional differences could be key factors in the differences observed between the free-living protein and the parasitic protein.

### 2.7. Hijacking ESPs for Research and a Cure

It follows logically from the observed skewing of the specific immune response to the Th2 phenotype that worms could be used as tools in order to interfere with the progression of clinical conditions associated with an aggressive Th1 immune response. This idea gained support in the frame of the formulation of the hygiene hypothesis [173,174,175]. Observations that originally led to the formulation of this hypothesis reported lower ratios of allergies and asthma in social environments associated with lower hygiene (Strachan). According to the hygiene hypothesis, infections—especially during childhood—can protect against allergic diseases. The original observation that a higher ratio of allergies and asthma is inversely correlated to worm infections has spurred interest in the use of nematodes and helminths in general as potential therapies to related conditions. Such observations have led to the suggestion of using helminths or helminth ESPs for the treatment of type 1 diabetes, inflammatory bowel disease and rheumatoid arthritis, as well as multiple sclerosis. This has been tested in animal models. In this review, we will illustrate only two examples of such animal models.

Metabolic syndrome is a clinical condition gathering abdominal obesity, a high plasma glucose concentration, dyslipidemia and hypertension. It is associated to a state of chronic inflammation and odds for type 2 diabetes and cardiovascular diseases. There are, indeed, several observations of correlations between worm infections and the improvement of some of the metabolic syndrome-associated parameters [176]. For example, an experimental infection with the filarial nematode *Litomosoides sigmodontis* improved the glucose tolerance in obese mice [177].

Atherosclerosis is a pathology-causing cardiovascular disease characterized by the thickening of the wall of large arteries. It is suggested to be triggered by lipid deposition in the intima and a sustained inflammation state. The experimental administration of helminth ESPs (SEA from *Shistosoma mansoni* (*T*) and ES62 from *Acanthocheilonema vitae* (*N*)) led to a reduction of circulating cholesterol and low density lipoprotein (LDL) levels and the size of atherosclerotic lesions in atherosclerosis-prone mice [178,179,180]. Furthermore, trials on small cohorts of patients suffering from Crohn’s disease or ulcerative colitis have been reported to improve after experimental infection with a pig whipworm, *Trichuris suis* (*N*) [181]. Several larger trials, however, did not conclude with any significant improvement of the clinical conditions associated with autoimmunity or allergy such as asthma, multiple sclerosis or coeliac disease [182]. Therefore, the use of helminths as therapies did not generate any conclusive benefits for patients with clinical conditions, despite reports of individual improvements, including in the community of self-treaters. There are, nevertheless, still ongoing trials using helminths as components of treatments for various diseases involving dysregulated immune responses: allergic diseases and autoimmune diseases. The only success recorded so far is for inflammatory bowel disease [183,184,185,186]. A major drawback of the use of worms for therapy is related to their pathological effects. One way to go around has been adopted, the use of controlled doses of worms responsible for benign infections. A second way to go is to use worms hosted by nonhuman species that are unable to reproduce inside the human body and disseminate eggs. The best way to go is likely a third, the use of ESPs. There have, indeed, been some reported positive effects of the administration of ESPs but, again, only in animal models.

Thus, for example, ES62 protected against neutrophil recruitment, hyper-bronchial inflammation and mucosal hyperplasia in a mouse model of airway hypersensitivity [187]. It also protected mice in a model of collagen-induced arthritis. In the same model, a MIF homolog of *Anisakis simplex* (*N*) ameliorated eosinophil infiltration and goblet cell hyperplasia [188]. It also protected against experimentally induced colitis in a mouse model [189]. Cystatin from *Acanthocheilonema vitae* (*N*) was reported to counteract inflammation [126] but, also, pollen-induced allergic responses in the lungs [190], as well as protect against experimentally induced colitis in a mouse model [126].

In summary, although there are encouraging individual reports, controlled reports in large trials for worm or ESP treatment are still awaited.

### 2.8. Available Tools for ESP Analysis

Most of the antigen characterization efforts in *Onchocerca* spp. have been geared towards the search for novel diagnostics, therapeutics and vaccines. This has been mainly based on screens using the sera of infected patients as tools. Limited efforts have been based on understanding the biology of the parasite, which could, nevertheless, help in the long term towards the same goals but, also, provide new knowledge on more fundamental subjects, such as potential mechanisms that may be specific to the parasite. Such knowledge may also help to address other broader issues outside of the scope of parasitology. As an example, the mechanisms by which nematodes heal the wounds they create during tissue invasion can help provide new drugs for tissue repair. Their ability to modulate the immune response would, of course, also be of great interest. The study of many nematodes suffers, unfortunately, from a major obstacle. They (or, at least, some of their parasitic stages) cannot be grown in the laboratory. In particular, it is still not possible to propagate *Onchocerca volvulus* in vitro. Modern approaches nevertheless contribute to make them more accessible to research.

#### 2.8.1. Genome Sequencing, Transcriptomics and Proteomics

Nematode genomics has improved drastically over the past decade with the sequencing of the genomes of an increasing number of species. Among those sequenced is the genome of *Onchocerca volvulus* [191], a total of 12,143 protein-coding genes were predicted in the *Onchocerca volvulus* genome, as deduced from the RNA-seq data from eight parasite stages. Stage-specific transcriptomics and proteomics analysis in *Onchocerca volvulus* revealed that over 75% of the genes had 100% transcript coverage in all stages, with the exception of the adult female stage, due to the fact that it is composed largely of uterine tissues containing other stages of the parasite [192]. Employing immunomic approaches in the study, a set of (sometimes new) proteins were identified as possible diagnostic markers for human onchocerciasis. The sequenced genome of *Onchocerca volvulus* still reveals about 44% of the proteins with no predicted function. Of the *Onchocerca volvulus*-specific genes, 92% encode putative proteins of unknown functions, of which 7% are potentially secreted [191]. To quickly identify potential antigens that could serve as vaccine, diagnostic and therapeutic targets, bioinformatics and proteomics approaches have also been applied. One example is a set of 33 diagnostic antigens proposed to serve as diagnostic markers for onchocerciasis [193]. Of these 33 diagnostic antigens, some are ESPs, but their individual diagnostic potentials are yet to be assessed.

Amongst the information of immediate use, the presence of proteins similar in sequence to human autoantigens was uncovered in the *Onchocerca volvulus* genome. Some of these autoantigens have been implicated in inducing cross-reactive antibodies that have been connected with the pathogenesis of posterior eye disease [194], as well as nodding syndrome [195]. *Onchocerca volvulus* genome variations in 27 isolates collected in the early 1990s prior to mass drug treatment from four endemic loci have been described [196]. In this study, discriminatory genomic regions between forest and savannah forms of onchocerciasis were uncovered, enabling the possibility to distinguish between forest, savannah and admixed strains. The evidence of unilateral gene flow from savannah to forest strains was also reported. High-throughput studies generate large amounts of data, a clear progress as compared to the “pre-omics era”, as they offer genes with their sequences, expression statuses and other characteristics for investigation. However, the specific confirmation of the involvement of each of these genes has to be confirmed, and their functions in *Onchocerca volvulus*/filariae/nematodes remain to be further analyzed. Therefore, there are, of course, many genes coding for ESPs that are potentially of interest in the frame of this review. This interest is sometimes supported by functions known in other organisms. We did not detail them, because nothing at all is known about them in *Onchocerca volvulus*. A few examples include acetylcholinesterase, chitinase or proteins involved in reactive oxygen species (ROS) detoxification.

#### 2.8.2. Animal Models

One central limitation in the functional characterization of nematode ESPs is the lack of suitable animal models of infection. While nematodes naturally infecting mice and other mammals have had a relatively easier path to the immunologic characterization of antigens, the same has not been true for parasitic nematodes of humans. On a general scale, a functional characterization in nematodes is mostly possible in *Caenorhabditis elegans*, which is a genetically tractable model. Thus, information could be obtained regarding the function of filariae genes through the analysis of their orthologs in *Caenorhabditis elegans* (*N*), where most genetic techniques (such as knock out (KO), clustered regularly interspersed short palindromic repeats (CRISPR)/Cas9 and transgenesis) are routinely applicable. However, a caveat to the use of *Caenorhabditis elegans* (*N*) is the fact that it is a free-living nematode, and parasitic inferences can be misleading. In parasitic nematodes, RNAi is possible in some life cycle stages of some species in which some parasitic stages can be maintained (but not expanded) in the laboratory. In *O. volvulus*, RNAi has been attempted using L3 worms [197,198], but it has been seldom used—certainly not as much as in the case in *Brugia malayi* (N), where many authors have reported its successful use [199,200,201,202,203,204]. For the purposes of drug screening in *Onchocerca volvulus*, the bovine parasite *O. ochengi* has been used as a model [205,206,207,208,209]. Comparative genomics revealed *Onchocerca ochengi* as the closest relative to *Onchocerca volvulus* [191], and the microfilariae that can be maintained a few hours in culture can be harvested from slaughterhouse biological samples in larger numbers. The *Onchocerca ochengi* model is, however, limited in its use for the functional characterization of *Onchocerca volvulus* ESPs due to its genetic intractability.

So far, in parasitic nematodes, the *Strongyloides species* is leading the way towards genetic tractability. Transgenesis has been reported in this species but, also, in *Brugia malayi* (*N*). CRISPR-Cas9 gene editing has also been successfully performed in *Strongyloides* (*N*) [210], which represents a crucial advancement that could be exploited for the functional and molecular characterizations of parasitic nematode ESPs.

## 3. Conclusions

As indicated in Table 1, there are only very few documented *Onchocerca volvulus* or, even, filariae ESPs and antigens. The small number probably reflects only a minor part of this group of molecules exposed and released by this multicellular and relatively complex organism. The fragmented information reported in this review also translates the restricted knowledge and, particularly, the imperfect understanding of their molecular actions, especially as compared to the importance of the sickness and the high potential of these molecules in their management either as diagnostic, curative or vaccine tools. The study of these molecules remains, therefore, a widely open field, with both vast fundamental interests and clinically applicable potentials.

## Figures and Tables

**Figure 1 pathogens-09-00975-f001:**
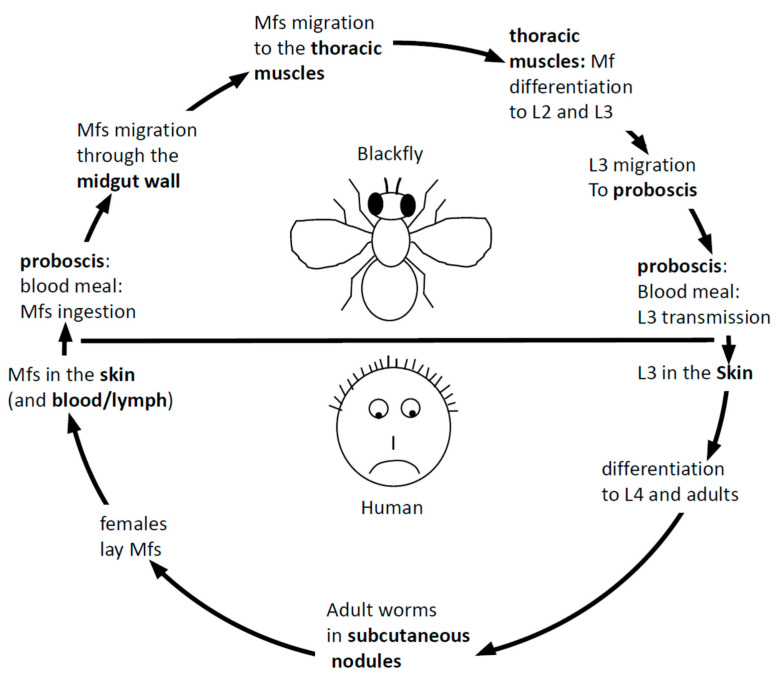
The parasitic cycle of *Onchocerca volvulus* (see text for full description). mf: microfilariae.

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
