# Peer review of "The Functional Parasitic Worm Secretome: Mapping the Place of Onchocerca volvulus Excretory Secretory Products"

_pathogens, 2020, doi:10.3390/pathogens9110975_

Round 1

Reviewer 1 Report

Overall comment:

The Review article by Vanhamme et al entitled “Nematode ESPs: Molecular and Functional Architecture” is well written and comprehensive. However, the lack of diagrams or figures makes it monotonous and wordy to lose the interest of readers.

Comments

  1. In the title, the acronym ‘ESPs’ can be avoided and may be replaced with its full form or secretome.
  2. In section, 1.3.1, a diagram would be better to explain the life cycle Onchocerca volvulus.
  3. In section 1.3.3, please make an illustrative diagram to explain the progression of pathogenesis.
  4. In section 2, line 222-238, Does the ESPs play an important role in communication with the host? What is the role of ESPs in phagocytosis of mf? Is there any elaborate study on the role ESPs on phagocytic process, if yes please cite those references? What are the chances of disease in immunocompromised versus healthy subject? what are the age groups highly vulnerable to get river blindness infections?
  1. Among ESPs biomolecules: proteins, glycoproteins, glycolipids and polysaccharides, what component(s) are known to act as a communicator with host efficiently to establish the infection? Please elaborate.

Author Response

Overall comment:

The Review article by Vanhamme et al entitled “Nematode ESPs: Molecular and Functional Architecture” is well written and comprehensive. However, the lack of diagrams or figures makes it monotonous and wordy to lose the interest of readers.

Comments

  1. In the title, the acronym ‘ESPs’ can be avoided and may be replaced with its full form or secretome.
  2. In section, 1.3.1, a diagram would be better to explain the life cycle Onchocerca volvulus.
  3. In section 1.3.3, please make an illustrative diagram to explain the progression of pathogenesis.
  4. In section 2, line 222-238, Does the ESPs play an important role in communication with the host? What is the role of ESPs in phagocytosis of mf? Is there any elaborate study on the role ESPs on phagocytic process, if yes please cite those references? What are the chances of disease in immunocompromised versus healthy subject? what are the age groups highly vulnerable to get river blindness infections?
  5. Among ESPs biomolecules: proteins, glycoproteins, glycolipids and polysaccharides, what component(s) are known to act as a communicator with host efficiently to establish the infection? Please elaborate.

Answers to Reviewer 1 comments

Point 1 : Done: ESP replaced by “excretory secretory products” in the title. The title has been changed in order to meet reviewer2 concerns

Point 2: A new Figure 1 now explains the O.v. parasitic cycle

Point 3: A. new Figure 2 now summarizes the progression of pathogenesis.

Point 4: Indeed, they play an essential role, for example in taming down the strength of the immune response. This is what we claim and detail in this review.

As far as phagocytosis is concerned, filariae interfere with the maturation of monocytes and dendritic cells as discussed in the review. They are able to trigger an alternative regulatory phenotype in these cell types that subsequently drive a TH2 response. As activation of DCs and macrophages is associated to phagocytic activity, an alternative activation would presumably come with perturbation of phagocytosis. This is supported by data from O’Regan et al. (Plos Negl.Trop.Dis. 2014, 8, e3206) indicating that Brugia malayi Mf lysates diminish monocyte phagocytic activity in cell culture (As these are crude lysates, an action of an ESP cannot be claimed at that stage). While alternative activation and a subsequent TH2 bias are clearly to the advantage of the worm, it is unclear whether suppression of phagocytosis is relevant as to the best of our knowledge worms are not phagocytosed (which seems logical as Mfs are at least 10 times larger than macrophages). There are old papers which claim that dead worm debris are eliminated by phagocytosis which seems logical.

We understand that because filariae interfere with and are fought by the immune response, it would be interesting to know what happens in cases of immunosuppression. Unfortunately, we are not aware of any study of the pathology of onchocerciasis or filariasis in immunocompromised patients. This point is also related to the next question of this reviewer, as ageing is associated to a decline of the immune response and a low level of inflammation.

However this evolution happens likely after 50, the age at which onchocerciasis pathology starts to be at its maximum and therefore age related immunodepression probably does not explain the evolution of the sickness. Thus  regarding possible age related effects, obviously, living in endemic regions daily enhances odds to be bitten by infected flies. This probably explains that, over the years, chances to be infected increase and pathology progresses in infected people. Accordingly, blindness and heavy skin symptoms are usually found in the elderly. Apart from this, regarding age-related effects, we only know about the nodding syndrome preferentially affecting children.

Point 5: While ESPs participating in the dialog with the host are detailed in this review, there are only very few cases where an active domain has been pinpointed in the molecules. The fucose group post translationally modifying LNFPIII and presumably involved in TLR4 binding or Phosphorylcholine addition to ES62 are such examples. A way to go to answer the reviewer’s question is obviously to undertake a deletional/mutational analysis of the molecules of interest but this is so far difficult as few nematode species are readily amenable to the necessary experiments. As mentioned in the paper there are alternatives: Take the opportunity of C elegans, an animal model where all the necessary techniques (transgenesis, CRISPR/Cas9, mutational analysis…) are routinely used. This is an approach we begun to attempt while studying GM2AP, again as said in the manuscript.  Another possibility is to produce mutated proteins in expression systems and use them in in vitro assays on cells of the immune system in culture for example.

Reviewer 2 Report

This manuscript from Vanhamme et al. is a very comprehensive review article focused on ESPs from different helminths. In my opinion it is very wide and authors have tried to encompass too much information from all different helminths and portray it in the manuscript, and, in its current form it is hard to follow since it jumps from nematodes to trematodes and back to O. volvulus and even C. elegans in some parts. In this regard, the title “Nematode ESPs: Molecular and Functional Architecture” doesn’t really reflect the aim and content of the manuscript, which is, essentially a review on the known functions of ESPs from too many helminths (not nematode in general). This is, in general, my main criticism, maybe authors want to re-focus the manuscript, change the title or both.

In addition, and this goes in relation to line 332, I am not sure “molecular architecture” is the right word in the title and throughout the text. As a reader, when I see molecular architecture I would think about domains present in the proteins, Pfam families to which ESPs belong, protein folding, etc… I think I understand what authors may want to say, but molecular biologists reading this work might get confused. I would strongly recommend to modify it to a more accurate title.

Sections 1.2 and 1.3 are perfect for a parasitology book, but I am not sure, if ESPs are the main focus of the paper, all this information needs to be part of the manuscript (except 1.3.4 and 1.3.6).

Line 62, there are 4 phyla of helminths, 2 of them (platyhelminthes and nematoda) important for humans, so I guess authors mean Genus?. Also, if the main focus of the manuscript are nematodes, why referring to Schistosoma and Taenia (lines 62-63) to highlight the importance of worms in terms of DALYs? Geohelminths also have a high impact on DALYs.

All paragraph from 265-311 talks about trematodes, not even nematodes and,no O. volvulus.

Reference 72 in line 259 doesn’t belong to any of the mentioned worms. Line 261: References such as 71 and Eichenberger et al (https://www.ncbi.nlm.nih.gov/pmc/articles/PMC5936971/) show that exosomes have a role in immunomodulation.

Lines 437, 503 and 504 (maybe others too) should say A. cantonensis.

Lines 443 and 457 (maybe others too), no capital A for Necator americanus

Throughout the text it should say N. brasiliensis (with “s”, not “z”)

Lines 534 and 535. Authors shoud put more attention to when worm names are cited for the first time (name in full) or not (shortened genus). C. elegans and O. volvulus has been mentioned before and should be shortened.

The manuscript is very thorough, and it capitulates a lot of the information we know about worm ESPs, but, as I said, I think it needs to be re-focused so it doesn’t look like it is jumping from one parasite to another. Maybe it should focus only on nematodes, or only on O. volvulus…

Author Response

Reviewer 2:

This manuscript from Vanhamme et al. is a very comprehensive review article focused on ESPs from different helminths. In my opinion it is very wide and authors have tried to encompass too much information from all different helminths and portray it in the manuscript, and, in its current form it is hard to follow since it jumps from nematodes to trematodes and back to O. volvulus and even C. elegans in some parts. In this regard, the title “Nematode ESPs: Molecular and Functional Architecture” doesn’t really reflect the aim and content of the manuscript, which is, essentially a review on the known functions of ESPs from too many helminths (not nematode in general). This is, in general, my main criticism, maybe authors want to re-focus the manuscript, change the title or both.

We partly agree with this view.

On one hand it is true that the paper navigates between different worm phyla and classes and we must admit that the information is very dense and that there is much information on non-nematodes (platyhelminths). This is related to our personal option to organize the information by ESP function and not by worm category. We also feel that we have limited ourselves to informations directly relevant to nematodes and Onchocerca volvulus. As there is really limited information on the pathological action on the O.v. secretome, we reasoned that this would be a nice complement.

On the other hand, we also limited ourselves very much while reviewing the ESPs, and focused on ESPs on which reliable functional information is available. We did not cover at all ESPs only recorded in the data banks as secreted proteins or putative ESPs…..for which function can only be extrapolated from known actions in other models (mammalian for example). For example we did not talk at all about proteins known in other systems to neutralize reactive oxygen species (superoxide dismutase, thioredoxin , or gluthathione-S-transferase), neurotransmitters  (acetylcholinesterase or aromatic amino-acid decarboxylase), collagenase, serpins, glycosyltransferases, tRNA synthetase or chitinase, all proteins proposed as nematodes ESPs.

Therefore, we followed the reviewer’s second proposal: change the title to a title that encompasses both worms in general and a focus on O.v.. Refocus the manuscript would involve  re-writing it, a work that would take more than the time allotted.

In addition, and this goes in relation to line 332, I am not sure “molecular architecture” is the right word in the title and throughout the text. As a reader, when I see molecular architecture I would think about domains present in the proteins, Pfam families to which ESPs belong, protein folding, etc… I think I understand what authors may want to say, but molecular biologists reading this work might get confused. I would strongly recommend to modify it to a more accurate title.

Agreed. We understand “molecular architecture” may be confusing. We changed it to molecular actions and functions throughout the text. The title has also been changed

Sections 1.2 and 1.3 are perfect for a parasitology book, but I am not sure, if ESPs are the main focus of the paper, all this information needs to be part of the manuscript (except 1.3.4 and 1.3.6).

This chapter was originally planned for a special issue dedicated to “Epilepsy related to Onchocerciasis” and was also designed to provide an introduction to Onchocerca, before to be redirected by the editors to another issue. We nevertheless believe that these informations are useful for the understanding of the described or postulated roles for the ESPs. If this referee maintains his/her request, we will of course remove those sections but we believe that they should then be replaced by a shorter text containing minimal information. On top of it, these sections are now illustrated by newly constructed figures as required by another reviewer. 

Line 62, there are 4 phyla of helminths, 2 of them (platyhelminthes and nematoda) important for humans, so I guess authors mean Genus?.

This has been corrected.

Also, if the main focus of the manuscript are nematodes, why referring to Schistosoma and Taenia (lines 62-63) to highlight the importance of worms in terms of DALYs? Geohelminths also have a high impact on DALYs.

This has been suppressed and replaced by a global DALY number encompassing all parasitic worms.

All paragraph from 265-311 talks about trematodes, not even nematodes and,no O. volvulus.

See general answer above

Reference 72 in line 259 doesn’t belong to any of the mentioned worms. Line 261: References such as 71 and Eichenberger et al (https://www.ncbi.nlm.nih.gov/pmc/articles/PMC5936971/) show that exosomes have a role in immunomodulation.

This has been dealt with and modified.

Lines 437, 503 and 504 (maybe others too) should say A. cantonensis.

Corrected

Lines 443 and 457 (maybe others too), no capital A for Necator americanus

Corrected

Throughout the text it should say N. brasiliensis (with “s”, not “z”)

Corrected

Lines 534 and 535. Authors shoud put more attention to when worm names are cited for the first time (name in full) or not (shortened genus). C. elegans and O. volvulus has been mentioned before and should be shortened.

Corrected. Furthermore, as we can understand that discrimination between the different genera, classes and phyla of worms can be a problem for readers, especially if many genera are cited, and that this can interfere with the understanding of the manuscript we have used throughout the text the full binomial system and indicated, each time into brackets the phylum or class of the considered worm.

The manuscript is very thorough, and it capitulates a lot of the information we know about worm ESPs, but, as I said, I think it needs to be re-focused so it doesn’t look like it is jumping from one parasite to another. Maybe it should focus only on nematodes, or only on O. volvulus…

See above